# Early Diagnosis of Fibromyalgia Using Surface-Enhanced Raman Spectroscopy Combined with Chemometrics

**DOI:** 10.3390/biomedicines12010133

**Published:** 2024-01-09

**Authors:** Haona Bao, Kevin V. Hackshaw, Silvia de Lamo Castellvi, Yalan Wu, Celeste Matos Gonzalez, Shreya Madhav Nuguri, Siyu Yao, Chelsea M. Goetzman, Zachary D. Schultz, Lianbo Yu, Rija Aziz, Michelle M. Osuna-Diaz, Katherine R. Sebastian, Monica M. Giusti, Luis Rodriguez-Saona

**Affiliations:** 1Department of Food Science and Technology, The Ohio State University, Columbus, OH 43210, USA; bao.172@buckeyemail.osu.edu (H.B.); delamocastellvi.1@osu.edu (S.d.L.C.); wuyalan5671@gmail.com (Y.W.); matosgonzalez.1@buckeyemail.osu.edu (C.M.G.); nuguri.2@buckeyemail.osu.edu (S.M.N.); siyuyao@seu.edu.cn (S.Y.); giusti.6@osu.edu (M.M.G.); rodriguez-saona.1@osu.edu (L.R.-S.); 2Department of Internal Medicine, Division of Rheumatology, Dell Medical School, The University of Texas, 1601 Trinity St., Austin, TX 78712, USA; 3Departament d’Enginyeria Química, Universitat Rovira i Virgili, Av. Països Catalans 26, 43007 Tarragona, Spain; 4Department of Nutrition and Food Hygiene, School of Public Health, Southeast University, Nanjing 210009, China; 5Department of Chemistry and Biochemistry, The Ohio State University, Columbus, OH 43210, USA; chelsea.goetzman@srnl.doe.gov (C.M.G.); schultz.133@osu.edu (Z.D.S.); 6Savannah River National Laboratory, Jackson, SC 29831, USA; 7Center of Biostatistics and Bioinformatics, The Ohio State University, Columbus, OH 43210, USA; lianbo.yu@osumc.edu; 8Department of Internal Medicine, Dell Medical School, The University of Texas, 1601 Trinity St., Austin, TX 78712, USA; rija.aziz@austin.utexas.edu (R.A.); michelle.osuna@austin.utexas.edu (M.M.O.-D.); kate.sebastian@austin.utexas.edu (K.R.S.)

**Keywords:** fibromyalgia, surface-enhanced Raman spectroscopy, central sensitization syndrome, metabolic fingerprinting, in-clinic disease diagnostics, chemometrics, blood

## Abstract

Fibromyalgia (FM) is a chronic muscle pain disorder that shares several clinical features with other related rheumatologic disorders. This study investigates the feasibility of using surface-enhanced Raman spectroscopy (SERS) with gold nanoparticles (AuNPs) as a fingerprinting approach to diagnose FM and other rheumatic diseases such as rheumatoid arthritis (RA), systemic lupus erythematosus (SLE), osteoarthritis (OA), and chronic low back pain (CLBP). Blood samples were obtained on protein saver cards from FM (*n* = 83), non-FM (*n* = 54), and healthy (NC, *n* = 9) subjects. A semi-permeable membrane filtration method was used to obtain low-molecular-weight fraction (LMF) serum of the blood samples. SERS measurement conditions were standardized to enhance the LMF signal. An OPLS-DA algorithm created using the spectral region 750 to 1720 cm^−1^ enabled the classification of the spectra into their corresponding FM and non-FM classes (Rcv > 0.99) with 100% accuracy, sensitivity, and specificity. The OPLS-DA regression plot indicated that spectral regions associated with amino acids were responsible for discrimination patterns and can be potentially used as spectral biomarkers to differentiate FM and other rheumatic diseases. This exploratory work suggests that the AuNP SERS method in combination with OPLS-DA analysis has great potential for the label-free diagnosis of FM.

## 1. Introduction

Fibromyalgia (FM) is a rheumatic disease of unknown etiology characterized by chronic widespread pain, sleep disturbances, physical exhaustion, autonomic dysfunction, and cognitive difficulties amongst many other symptoms making it difficult to diagnose [1]. Estimates of the prevalence of FM have been made in various settings, multiple countries and across five continents: Africa, North and South America, Asia, and Europe. The global mean prevalence of FM was 2.7% [2], with estimates ranging from a low of 0.4% in Greece to a high of 9.3% in Tunisia [3,4]. In North and South America, the average rate is approximately 3.1%, 2.5% in Europe, and 1.7% in Asia. Prevalence rates globally in women and men are 4.2% and 1.4%, respectively, with a female-to-male ratio of 3:1 [2]. Currently, there are no reliable diagnostic tests; thus, historical intake coupled with careful clinical evaluative descriptions are used to identify affected individuals. Fibromyalgia is frequently associated with other rheumatic diseases such as rheumatoid arthritis (RA), systemic lupus erythematosus (SLE), osteoarthritis (OA), chronic low back pain (CLBP), and many other chronic pain disorders [5,6,7]. Physicians are in need of a rapid diagnostic method to easily discriminate between fibromyalgia and other rheumatic conditions [8]. Our research group has been a pioneer in using vibrational spectroscopy technology to clearly distinguish FM patients from RA and OA groups using pattern recognition software with 100% accuracy and no misclassifications [9]. These results have been further confirmed by Passos et al. [10]. These researchers created an algorithm with linear discriminant analysis to discriminate FM and healthy patients using mid-infrared spectral data. Amide II (1545 cm^−1^) and proteins (1425 cm^−1^) were also identified to be discriminant features. Discovery of a reliable biomarker for FM would be a critical step towards early intervention [11]. A definitive diagnosis of FM often takes many years, with innumerable clinic visits, investigations, and specialist consultations [12]. Our group has also reported the first metabolomics studies to diagnose FM and related rheumatologic disorders RA, SLE, and OA using vibrational mid spectroscopy [13]. Aromatic amino acids and peptide backbones have been highlighted as potential biomarkers for FM [14]. More recently, the deconvolution analysis applied to spectral data of FM and long COVID patients has led us to identify a unique spectral band at 1565 cm^−1^ linked to stretching vibrations of carboxylate groups of amino acid side chains only present in FM patients [15].

Surface-enhanced Raman spectroscopy (SERS) is a non-invasive method currently used in clinical diagnosis due to its sensitivity in the study of biological samples (i.e., blood, serum, and plasma). SERS provides unique biomarker information present at very low concentrations in a physiological environment [16]. The detection limit in SERS ranges mostly from ng/mL to fg/mL [17]. SERS signal enhancement relies on two enhancement mechanisms: electromagnetic enhancement (EE), originating from the excitation of localized surface plasmon resonance (LSPR) in metal nanoparticles (NPs), and chemical enhancement (CE), produced by chemical interactions between the surface on the metallic NPs and the analyte. SERS uses plasmonic NPs to enhance the Raman signal by factors up to 10^6^ [18]. The size, size distribution, shape, and chemical composition of metallic NPs play a dominant role [19]. Gold, silver, and copper are normally used to synthetize these NPs as their corresponding plasmon resonances occur near the visible range of light [20]. Among them, AuNPs are less susceptible to oxidation, are more biocompatible, and show a strong plasmon excitation at 785 nm [21]. Moreover, AuNPs exhibit a high affinity for metabolites, nucleic acids, and proteins because of their highly electronegative or charged atoms [22]. Some vibrational modes’ energy, including that of C–C and N–N bands, is especially important because it is not visible with infrared spectroscopy [23]. The SERS method requires a small amount of sample with minimal sample preparation and has low water interference, allowing the analysis of human fluids in a rapid, real-time, and non-destructive way. For instance, by using AgNPs on serum, SERS spectral data combined with Linear Discriminant Analysis (LDA) has been used to successfully discriminate patients with Chagas disease from those that were healthy and asymptomatic [24]. Moreover, SERS has been used in the diagnosis of chronic kidney diseases using AgNPs and serum samples [25]. As far as we know, no previous research has investigated the potential of using SERS in the low-molecular-weight fraction (LMF) of the human plasma proteome to diagnose FM and related rheumatologic disorders. Thus, we hypothesize that vibrational spectroscopy may provide a powerful tool for differentiating FM from other disorders. The additional capabilities of SERS may add multifold capabilities to this detection ability. These studies may lead to evidence of prospective therapeutic targets for FM-associated pain.

The objective of this research was to use the SERS spectroscopic technique combined with Orthogonal signal correction–partial least squares discriminant analysis (OPLS-DA) for the classification of the patients into two groups, FM and other rheumatoid diseases (RA, SLE, OA, and CLBP). In addition, unique spectral fingerprints for FM, RA, SLE, OA, CLBP, and healthy patients (NC) were detected.

## 2. Materials and Methods

### 2.1. Patient Sample Recruitment and Sample Storage

Approval from the University of Texas at Austin institutional review board was obtained prior to embarking on any studies with human subjects. Clinical registries focus on cataloguing the efficacy of medical interventions in clinical trials to achieve health endpoints. As there were no medical interventions conducted during this study, it did not meet the accepted National Institutes of Health criteria for a clinical trial. Thus, it is not included in an open registry. All studies adhered to Declaration of Helsinki principles. The IRB approval date was 19 June 2020 (study no. 2020030008). Following informed consent, blood samples were obtained from patients with FM (*n* = 83), and those with other rheumatic disorders (RA, SLE, OA, CLBP, *n* = 54) and healthy controls (NC, *n* = 9), at the University of Texas at Austin clinics and the Dell Seton Ascension Rheumatology Clinics in Austin, Texas. Venous blood was collected in ethylenediamine tetra acetic acid (EDTA)-laced tubes. Bloodspot aliquots from the tubes for all subjects were obtained between September 2020 through June 2023. Samples were collected and stored on bloodspot cards (Whatman 903 Blood Protein Saver Snap Apart Card, GE Healthcare, Westborough, MA, USA) at −20 °C until they were shipped to the Rodriguez-Saona Vibrational Spectroscopy laboratory at The Ohio State University Department of Food Sciences on dry ice and stored at −20 °C until subsequent extraction and analysis. Standardized circles on the filter paper served as a guide to ensure collection of approximately 50 µL of blood per spot.

Questionnaires: All subjects provided a self-report of symptoms through use of the Revised Fibromyalgia Impact Questionnaire (FIQR), a 10-item self-rating instrument that measures physical functioning, work status, depression, anxiety, sleep, pain, stiffness, fatigue, and wellbeing. The Beck Depression Inventory (BDI) is a 21-item, self-report rating inventory that measures characteristic attitudes and symptoms of depression [26]. The Revised Symptom Impact Questionnaire (SIQR) is the FM-neutral version of the FIQR and does not assume the patient has FM. The SIQR was utilized as a measure of physical functioning, work status, depression, anxiety, sleep, pain, stiffness, fatigue, and wellbeing on all subjects without FM and normal controls [27,28,29]. The Central Sensitization Inventory (CSI) is a two-part patient-reported outcome measure that assesses somatic and emotional symptoms common to CSS [30]. The McGill Pain questionnaire (MPQ) is an instrument providing descriptive aspects of pain as well as pain intensity [31].

Criteria for the diagnosis of FM included: age 18–80 with a history of FM and meeting current criteria for FM [32]. Criteria for diagnosis of osteoarthritis (OA) included subjects with age 18–80 with morning stiffness < 30 min in duration, crepitus, and radiographic evidence of OA or clinician confirmation with lack of evidence of a concurrent inflammatory component. Chronic low back pain subjects’ inclusion criteria were age 18–80, low back pain for at least 3 months, and meeting the criteria of the American Pain Society [33]. Systemic lupus erythematosus inclusion criteria were subjects ≥18 years with defined SLE according to the revised ACR/EULAR classification criteria [34]. Rheumatoid arthritis (RA) inclusion criteria were age 18–80 and meeting ACR/EULAR criteria for rheumatoid arthritis [35]. Sigmaplot v15.0 and SigmaStat v4.0 software (Inpixon, Palo Alto, CA, USA) were utilized for statistical analysis of questionnaires. A flow diagram of the study is displayed in Figure 1.

### 2.2. Blood Sample Preparation

A low-molecular-weight fraction (LMF) of the human blood sample was obtained following the protocol [14] with minor modifications (see Appendix A). A bloodspot circle (50 µL of blood) was cut in sterile conditions from a Whatman^TM^ 903 protein saver bloodspot card (GE Healthcare, Westborough, MA, USA and placed in a 15 mL centrifuge tube with 1 mL of autoclaved HPLC grade water (Sigma-Aldrich, St. Louis, MO, USA). Subsequently, the blood solution was sonicated (Sonic Dismembrator Model 100, Fisher Scientific, Pittsburgh, PA, USA) for 30 min. The resulting dissolved blood aliquot was subjected to a filtration process using an Amicon^®^ ultra-centrifugal filter membrane tube (Sigma-Aldrich, St. Louis, MO, USA). This tube was previously rinsed with 3 mL of HPLC grade water and centrifuged (Sorvall™ Legend™ XFR Centrifuge, Thermo Fisher Scientific Inc., Waltham, MA, USA) at 4000 rpm for 10 min at 4 °C. This procedure was repeated four times to remove the glycerol that coated on the filter membrane [14,36]. Then, the dissolved blood aliquot was transferred to the washed filter tube and centrifuged at 4000 rpm for 15 min at 4 °C to obtain the low-molecular-weight fraction (LMF) of the human plasma proteome. Water was completely removed from the LMF supernatant, first by using a nitrogen gas concentrator (BTLab 103 Systems, BenchTop Lab System, St. Louis, MO, USA) until the volume was reduced to 0.5 mL and then by vacuum centrifuging (Vacufuge plus Concentrator, Eppendorf, Westbury, NY, USA). The dried samples were kept in the freezer for further analysis.

### 2.3. AuNP Preparation and Characterization

HAuCl_4_·H_2_O (CAS 254169) and sodium citrate tribasic dihydrate (CAS 6132-04-3) were purchased from Sigma-Aldrich (St. Louis, MO, USA). Milli-Q water was used for all solution preparations and experiments. AuNPs were synthesized using the citrate reduction method [37] following the protocol from Zoltwski et al. [38]. Briefly, 0.061 g HAuCl_4_ was mixed with water (500 mL) and mixed (350 rpm) and heated to 90 °C for 30 min. After that, a sodium citrate solution (7.5 mL, 0.08%) was rapidly added to the heated solution under stirring conditions. After 3 min, the resulting AuNP dispersion, whose color changed from yellow to red, indicating the formation of monodisperse spherical particles, was cooled to room temperature. The obtained AuNPs were stored in glass vials (Restek, Centre County, PA, USA) at room temperature in the dark until further use.

AuNPs were characterized via UV–visible absorption spectroscopy after each preparation and use using an Agilent 8453 UV−Vis spectrometer (Agilent Technologies, Inc., Santa Clara, CA, USA). The average extinction band maximum was 538 nm (see Appendix A, for further details). Dynamic light scattering (DLS) was performed to measure particle size using a Malvern Zetasizer system (Malvern Panalytical Ltd., Westborough, MA, USA). The average particle size was 36 nm (see Appendix A, for further details).

Prior to the SERS analysis (Figure 2), 1 mL of AuNPs was centrifuged at 6000 rpm for 25 min to remove the supernatant. The pellet was then diluted with Milli-Q water and acetonitrile at a ratio of 1:1 (*v*/*v*) to a final volume of 1 mL. SERS samples were prepared by vertexing 10 µL AuNPs with the dried LMF sample for 10 s using a Mini Vortex Mixer (Thermo Fisher Scientific Inc., Waltham, MA, USA) and centrifuging at mySPIN™ 6 mini centrifuge, (Thermo Fisher Scientific Inc., Waltham, MA, USA) for 10 s. Then, 5 µL of the mixture of LMF and AuNPs was transferred onto an aluminum-covered well slide (BRAND company Inc., South Hamilton, MA, USA) for the SERS analysis.

### 2.4. SERS Equipment and Measurement

The SERS spectra were acquired with an Optical PhotoThermal InfraRed (OPTIR) Raman micro-spectrometer (Photothermal Spectroscopy Corp, Santa Barbara, CA, USA). Excitation was obtained using a 785 nm diode laser with an output power of 97 mW. The laser light was passed through a line filter and focused on a sample mounted on an X–Y–Z translation stage with a 50× objective lens (numerical aperture 0.8) that focused the laser to a spot size of 0.6 μm with a spectral range of 600 and 2300 cm^−1^. A total of 600 grooves per mm grating was used to provide a spectral resolution of 2 cm^−1^. The SERS spectra were acquired for 20 s, with 8.7 mW of the laser power measured at the sample with 1 scan. Each sample was analyzed four times, and the average result was used as the sample spectral data. Calibration was checked using the 520 cm^−1^ vibrational band of a silicon wafer as a reference. Spectral acquisition was controlled by PTIR studio software (version 4.5.1, Photothermal Spectroscopy Corp, Santa Barbara, CA, USA).

### 2.5. Data Processing and Chemometrics Analysis

The autofluorescence background removal was performed using a Rubberband baseline correction algorithm with the PTIR studio software (Version 4.5.1, Photothermal Spectroscopy Corp). Then, all background-subtracted SERS spectra were imported into chemometrics analysis software, Pirouette^®^ version 4.5 (Infometrix Inc., Woodville, WA, USA). Next, spectra were mean-centered, and the Savitsky–Golay (SG) second derivative (25 points) was applied [39]. Orthogonal signal correction–partial least squares discriminant analysis (OPLS-DA) was used to discriminate FM samples from other rheumatic disorders (RA, SLE, OA and CLBD). OPLS-DA is a supervised learning technique which calculates a regression relationship between Raman data and a response variable that contains known class memberships. The orthogonal signal correction (OSC) approach operates by identifying and removing the uncorrelated data from the X to the Y matrix to minimize the variance between individuals.

The dataset was divided (Table 1) to train the algorithm and subsequently assess its performance using an independent test set. A total of 80% of the dataset was used to build the calibration model (*n* = 68 FM and *n* = 41 other rheumatic diseases), and the remaining 20% was used to externally validate of the dataset (*n* = 15 FM and *n* = 13 other rheumatic diseases).

Internal cross-validation (ICV) of the calibration model was conducted employing a leave-one-out approach, where each sample was systematically excluded in turn to develop a model predicting class membership. That remaining sample were then employed to evaluate the discriminatory capacity of the OPLS-DA model [40]. The internal cross validation provided a performance estimate of the calibration model including correlation coefficient of cross validation (R^2^cv) and standard error of cross validation (SECV), while external validation (EV) with an unseen dataset revealed the correlation coefficient of prediction (R^2^pre) and standard error of prediction (SEP) and presented the model’s performance when deployed in real-world scenarios for FM diagnosis [41]. The EV assessment result provides the predictive accuracy, sensitivity, and specificity of the model. The ROC plot of external predictions was computed using the pROC package [42] in R software (Version 4.3.1) [43]. The ROC plot evaluates the performance of the diagnostic tool at all possible thresh olds and provides the area under the curve (AUC), which assesses the model’s accuracy [44]. A higher AUC indicates better accuracy.

## 3. Results

### 3.1. Clinical Characteristics of Subjects

The clinical characteristics of all subjects are presented in Table 2 and Table 3. Table 2 shows 83 subjects with 7 males and 76 females. They had a mean age of 42.2 ± 14.1 Their BMI was 30.9 ± 8.3 with a CSI of 64.6 ± 15.1. Their FIQR was 54.6 ± 21.4, MPI was 100.3 ± 48.9, and BDI was 23.5 ± 11.1. Subjects with other rheumatic disorders are also shown (non-FM numbered 54 with 10 male and 44 females). They had a mean age of 52.2 ± 16.4. Their BMI was 31.2 ± 12.8 with a CSI of 35.4 ± 16.3. Their SIQR was 33.5 ± 23.0, MPI was 44.3 ± 39.3 and BDI was 9.5 ± 8.6. Healthy control subjects numbered 9 (5 males and 4 females) with a mean age of 45.0 ± 17.7. Their BMI was 25.6 ± 4.5 with a CSI of 15.8 ± 12.8. Their SIQR was 4.3 ± 7.0, MPI was 5.3 ± 8.9, and BDI was 0.7 ± 0.8.

Table 3 is a sub-analysis of the non-FM group. RA subjects numbered 23 with 4 males and 18 females. Their mean age was 51.44 ± 15.55. Their BMI was 29.1 ± 9.7 with a CSI of 35.4 ± 14.4. The SIQR was 37.3 ± 2 3.8, MPI was 42.3 ± 34.6, and the BDI was 9.4 ± 7.4. SLE subjects numbered 17 with 2 males and 15 females. Their mean age was 43.67 ± 15.3. Their BMI was 33.65 ± 9.6 with a CSI of 33.27 ± 19.6. The SIQR was 31.2 ± 25.5, MPI was 51.1 ± 53.5, and the BDI was 10.4 ± 11.0. OA subjects numbered 12 with 3 male and 9 females. Their mean age was 67.1 ± 8.9. Their BMI was 30.7 ± 10.2 with CSI of 36.8 ± 15.5. Their SIQR was 25.4 ± 12.2, MPI was 54.6 ± 36.6, and the BDI was 7.3 ± 5.4. CLBP subjects numbered 2 with 1 male and 1 female. The mean age was 59.5 with a BMI of 53.4, CSI of 53, SIQR of 65.2, MPI of 80, and BDI of 15.

Table 4 displays a statistical comparison of FM and non-FM survey scores with regard to age, BMI, CSI, FIQR/SIQR, MPI, and BDI. *p*-values represent two-tailed comparisons between groups. Statistically significant differences between the FM and non-FM groups were seen for all measures at the <0.005 level except for BMI.

### 3.2. Effect of AuNPs on the Signal Enhancement of LMF

AuNP SERS and regular Raman spectra of the same FM patient are presented in Figure 3. The spectral intensity significantly increased in major vibration bands in the SERS signal (Figure 3, a), suggesting strong bindings between biomolecules in the LMF sample and AuNPs. On the other hand, no defined peaks appeared in the regular Raman spectrum of the analyzed FM subject (Figure 3, c), which confirmed the localized surface plasmon resonance enhancement effect of AuNPs on LMF blood samples [45]. Moreover, the plasmonic enhancement phenomenon is also sensitive to the composition of AuNP solution. The solvent used to dilute AuNP pellets influences the enhancement of the Raman signal. Our preliminary data showed that when the AuNP pellet was diluted with Milli-Q water, low SERS spectral signals were achieved (Figure 3, b). This is due to H_2_O, as a solvent, having low solubility of many organic reagents in electrochemical transformations [46]. AuNP pellets diluted with H_2_O/ACN at a 1:1 ratio gave significantly higher SERS spectral signals (Figure 3, a). Acetonitrile provides a broader range of solvent polarities and dielectric functions in the medium surrounding plasmonic materials [46]. Finally, the AuNP solution did not have SERS spectral signals by (Figure 3, d), which eliminated the potential influence of AuNP solution on the separation of FM vs. non-FM classes in the chemometric analysis.

### 3.3. SERS Measurement Conditions

Four laser power intensities, 2.1, 4.3, 8.7, and 16.5 mV, were evaluated with 20 s acquisition time (Figure 4a). A low laser power of 2.1 mV demonstrated weak Raman signals, while higher laser powers (4.3 and 8.7 mV) gave consistent signals. In addition, sharper peaks with higher intensities were observed with 8.7 mV laser power. Increasing the laser power to 16.5 mV revealed an additional band at 1535 cm^−1^ with inconsistencies in the spectral signals. Figure 4b demonstrates the characteristic band associated with the photodegradation of the substrate due to high heat accumulation from 16.5 mV laser power [32,33,34,35]. To eliminate these artifact signals, 8.7 mV laser power was determined to be appropriate for sample measurements.

Once the optimal laser power was selected (8.7 mV), four different acquisition times, 5, 10, 15, and 20 s, were tested (Figure 5). SERS signal intensities decreased with the acquisition time with no valuable spectral information at 5 s. On the contrary, when the exposure time was increased to 20 s, defined Raman peaks were obtained without the indicator (1535 cm^−1^) of a burned signal. Therefore, 20 s was the exposure time selected for analysis of our LMF blood samples.

### 3.4. SERS

The normalized and averaged SERS spectra of LMF blood samples from FM and other rheumatic disease (OA, RA, SLE, and CLBP) subjects are presented in Figure 6a. Three dominant regions were detected. Between 1100 and 1500 cm^−1^, five intense bands, 1129, 1189, 1222, 1354, and 1448 cm^−1^, produced by C-H bending (1129 and 1189 cm^−1^); amide III or C-H stretching (1222 cm^−1^); and CH_2_, CH_3_ bending (1354 and 1448 cm^−1^), were observed. Within this zone, amino acids such as tryptophan (Trp), phenylalanine (Phe), and tyrosine (Tyr) as well as phospholipids and lipids can be absorbed (Table 5). The area between 900 and 1050 cm^−1^ contained bands 911, 967, and 1014 cm^−1^ linked to C-C stretching of lysine (Lys) and Phe and benzene ring breathing of Trp. Finally, the area between 1550 and 1650 cm^−1^ had bands (1586 and 1633 cm^−1^) related to the C=C double bond of Phe, Tyr, and the beta sheet of amide I. The second derivative (SG 25) was applied to the normalized and averaged SERS spectra of LMF blood samples to emphasize band widths, position, and separations (Figure 6b). The highest intensity differences between diseases were detected at 1014, 1222, 1453, and 1586 cm^−1^. Once more, these bands were mostly associated with plasma amino acids such as Phe, Tyr, and Trp.

Figure 7 shows the normalized and averaged SERS spectra of LMF blood samples from NC subjects and FM patients. We subtracted the normalized and averaged FM spectra from the NC to detect intensity changes in the SERS signals. Band intensity changes appears in five major bands, 1018 and 1222 cm^−1^, attributed to C-H bending of Tyr, Phe, and amide III; 1354 cm^−1^, corresponding to CH_2_ bending of Trp; 1453 cm^−1^, linked to CH_2_ and CH_3_ bending of phospholipids and lipids; and finally 1586 cm^−1^, related to C=C bonding of Phe and Tyr.

To evaluate the ability of LMF SERS spectra to effectively distinguish between FM and other rheumatic diseases, we performed OPLS-DA analysis on 80% of the dataset (FM *n* = 68 as class 1, and non-FM *n* = 41 as class 2). OPLS-DA was used to analyze the covariance pattern of spectral variables (X) and disease response (Y) and enhanced the PLS algorithm’s predictability by eliminating orthogonal variation in X not describing Y [x]. The Y matrix included binary labels indicating FM and non-FM categories. The parameters, specifically the spectral region and pre-processing methods, were optimized by reconstructing the model under various conditions. The OPLS-DA algorithm developed using the 750 to 1750 cm^−1^ region provided the best performance. The X variables were mean-centered, normalized, and second-derivatized (SC, 25) with the use of one orthogonal signal correction (OSC) component, which made the predictive quality of the model satisfactory.

The OPLS-DA model showed figures of merit (R^2^ and SECV/SEP) that were compatible with high performance (Table 6) and a low *p*-value (*p* < 0.05), suggesting significant discrimination and significant differences between FM and other rheumatic subjects (RA, OA SLE, CLBP).

The score plot of OPLS-DA regression models obtained from spectral data is presented in Figure 8a. The score plot showed distinctive clusters of spectra from subjects with FM and other rheumatic diseases, and seven latent variables (LVs) were needed to explain over 70% of our algorithm variation (with the initial three LVs explaining 48.8%). Fewer than 3% of the data points were found to be influential with a high leverage and studentized residual; they were eliminated to improve the predictive accuracy of the model. The classification performance of this algorithm was tested using an independent set of samples (*n* = 15 FM and *n* = 13 non-FM), providing excellent unbiased predictions with no misclassification, indicating 100% accuracy, sensitivity, and specificity performance (Table 6). These results show that the OPLS-DA algorithm built with SERS spectral data can be used for screening of FM subjects.

Regression vector (Figure 8b) analysis showed that the discriminating region was dominated by 991, 1013, 1202, 1222, 1348, 1370, 1566, and 1582 cm^−1^ bands mainly linked to plasma amino acids such as Try, Phe, Ser, and Trp. The detailed vibrational modes with assigned biomolecular components are summarized in Table 7 with references.

The ROC plot exhibits the sensitivity and specificity of the model’s response to all possible thresholds that define the result as positive [33]. The ROC plot gave a perfect AUC of 1, indicating excellent classification (Figure 9) [56].

## 4. Discussion

Fibromyalgia (FM) is a central sensitivity disorder characterized by widespread muscle pain. Additionally, patients with FM commonly experience symptoms including depression, anxiety, fatigue, memory, and sleep disturbances [57]. Due to the presence of overlapping symptoms with other rheumatic diseases, FM is often considered a diagnosis made after excluding multiple other alternative conditions. The work involved in leading to a confirmatory diagnosis may take on average approximately 3–5 years of diagnostic testing for individuals with FM [58]. This gap in research is particularly crucial, given that the difficulty in differentiating FM from other rheumatic diseases contributes significantly to the prolonged diagnosis period for FM. Our comparator groups of FM and non-FM patients were closely matched with regard to BMI. The statistically significant differences in FIQR/SIQR, BDI, CSI, and MPI between these two groups were expected given the differences in clinical presentation between these conditions. In addition, there is a predilection for individuals with FM to display increased features of central sensitization; characteristics of which are picked up in all of the survey elements but most prominently by the CSI where values on the CSI of >40 are consistent with central sensitization syndrome (CSS) [30].

Surface-enhanced Raman spectroscopy (SERS) uses the excitement of the localized surface plasmon resonance of AuNPs to enhance local electric fields. When biomolecules adsorb onto AuNPs, the electromagnetic field around them is enhanced, leading to an increase in the Raman signal of the adsorbed molecules [59]. However, biomolecules in LMF blood samples typically exhibit low scattering properties and high sensitivity to radiation damage [60]. To increase the scattering properties, we modified the AuNP solution to increase the binding affinity between AuNPs and biomolecules in the LMF blood samples [61]. We found that AuNPs diluted with water alone yielded spectra with lower intensity due to the limitation of water’s low solubility of many organic reagents in electrochemical transformations [46]. When acetonitrile to water at a 1:1 ratio (*v*/*v*) was used to dissolve AuNPs, a higher spectral intensity was obtained. Acetonitrile has a broader range of solvent polarities and dielectric functions in the medium surrounding plasmonic materials, leading to an increase in spectral intensity [46]. Biomolecules in our LMF blood samples may be sensitive to radiation damage. SERS electromagnetic radiation can produce photodegradation in the sample, which was determined to be caused by higher-energy laser exposure or prolonged exposure times through photothermal, photomechanical, and photochemical mechanisms [61,62]. To avoid damaging them, we optimized the SERS measurement conditions by testing different laser powers and acquisition times. When our LMF blood samples suffered photodegradation, the band at 1535 cm^−1^ associated with amorphous carbon was generated as a result of damage to the AuNP substrate from high-power laser excitation [63,64,65].

The normalized SERS spectra of FM subjects were very similar to those of non-FM disease subjects. Thus, it was necessary to apply multivariate analysis to obtain more information from the SERS spectra. OPLS-DA is commonly used to handle highly collinear and noisy data that are commonly output in metabolomics experiments [52]. In this study, OPLS-DA was used to develop an algorithm to discriminate FM patients from RA, OA, SLE, and CLBP patients. The OPLS-DA predictive algorithm successfully distinguished individuals with FM subjects and other rheumatic diseases with no misclassification in either the calibration model or external validation model, demonstrating the capability of using label free AuNP SERS in early diagnosis of FM. Furthermore, the accuracy of the generated algorithm was classified as excellent based on the AUC value computed in the ROC plot [56], which proved the model’s performance when deployed in real-world scenarios for FM diagnosis. Raman bands at 991, 1013, 1202, 1222, 1348, 1370, 1566, and 1582 cm^−1^, which are associated with plasma amino acids Phe, Tyr, and Trp, were highlighted in the regression vector plot to discriminate diseases using the OPLS-DA algorithm. These results are in agreement with previous studies performed by our research group using mid-infrared spectroscopy. Aromatic amino acids and peptide backbones were the biomolecules that most effectively discriminated FM from other rheumatic diseases [14,66]. It has also been reported that FM patients have abnormal plasma amino acid levels such as phenylalanine, proline, glycine, lysine, tryptophan, and tyrosine, which are related to human neuronal functioning [53,67,68]. These outcomes support the notion that plasma amino acids serve as a potential biomarker for diagnosis of FM.

Our current study is ongoing and aims at metabolically differentiating FM from other rhematic diseases through AuNP SERS. Building upon these promising outcomes, our future research will dive deeper into identifying key biomarkers associated with FM, thereby contributing valuable insights to the understanding and early detection of this disease.

## 5. Conclusions

In conclusion, we have developed a standard operating procedure (SOP) for measuring LMF blood samples using AuNP SERS. The OPLS-DA algorithm was able to accurately classify the subject based on their disease type (FM vs. non-FM). All subjects were assigned to their respective classes in both the internal cross-validation (ICV) calibration model and the external validation set with no misclassification and a diagnostic accuracy, sensitivity, and specificity of 100%, which reflect the trustworthiness of the prediction model. The results prove that AuNP SERS coupled with chemometrics can be used as a real-time point-of-care device for screening of FM. 

## Figures and Tables

**Figure 1 biomedicines-12-00133-f001:**
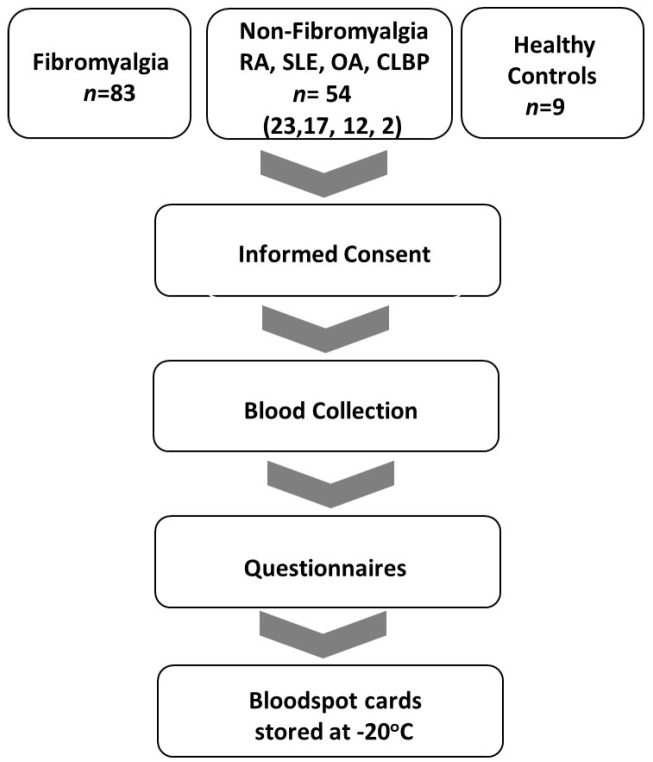
Study flow diagram. Inclusion of FM, non-FM (RA, SLE, OA and CLBD), and NC patients in the period from September 2020 to June 2023.

**Figure 2 biomedicines-12-00133-f002:**
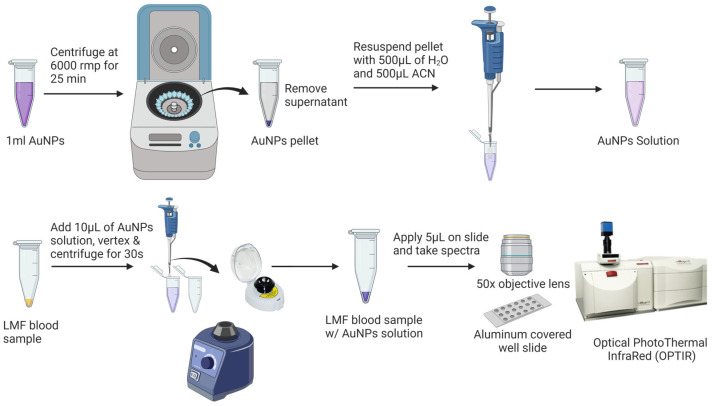
Low-molecular-weight fraction (LMF) blood sample preparation for SERS analysis.

**Figure 3 biomedicines-12-00133-f003:**
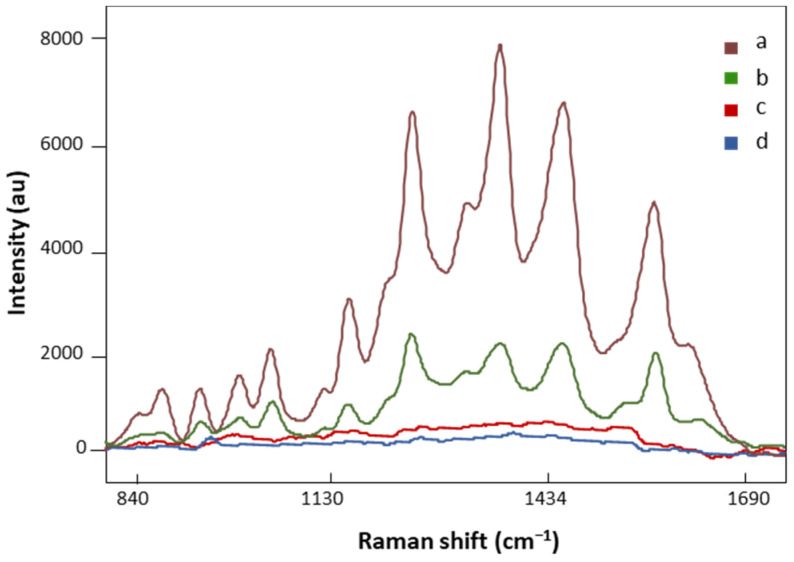
The SERS spectra of LMF blood samples from the same FM subject with (a) H_2_O and ACN at a 1:1 ratio diluted AuNP pellet, (b) Milli-Q water diluted AuNP pellet; (c) the regular Raman spectrum of LMF blood samples without AuNP solution; (d) the Raman spectrum of AuNP solution alone. All spectra were obtained with 8.7 mV power and 20 s.

**Figure 4 biomedicines-12-00133-f004:**
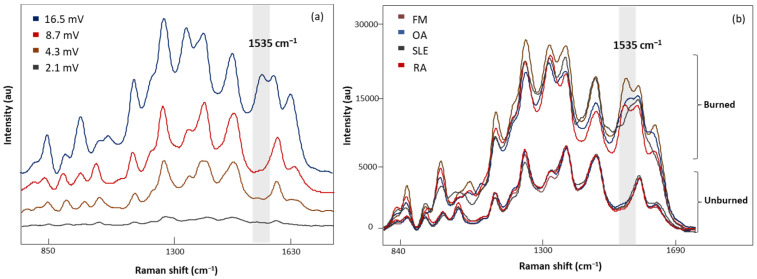
(**a**) The SERS spectra of LMF blood samples from the same FM subject measured at different laser powers, 2.1, 4.3, 8.7, and 16.5 mV, for 20 s, (**b**) average SERS spectra of unburned and burned (photodegraded) sample from FM OA, SLE, and RA subjects, respectively. Grey region shows the Raman band linked to the burning phenomena.

**Figure 5 biomedicines-12-00133-f005:**
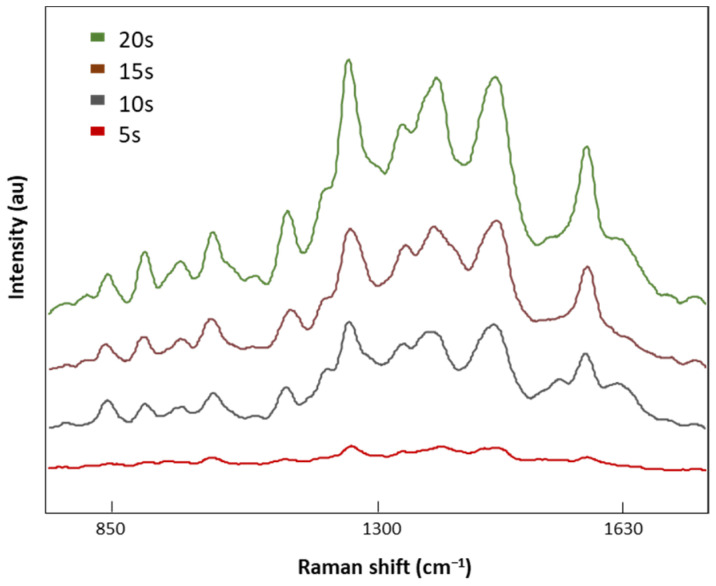
The SERS spectra of LMF blood samples from the FM subject measured with different acquisition times of 5, 10, 15, and 20 s at 8.7 mV.

**Figure 6 biomedicines-12-00133-f006:**
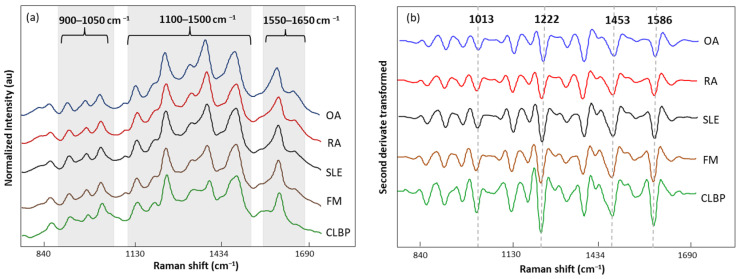
(**a**) Normalized and averaged SERS spectra of LMF blood samples from FM (*n* = 83) and non-FM disease (*n* = 54) subjects; (**b**) second derivative (SG 25) transformed and averaged SERS spectra of LMF blood samples from FM (*n* = 83) and non-FM diseases (*n* = 54) subjects.

**Figure 7 biomedicines-12-00133-f007:**
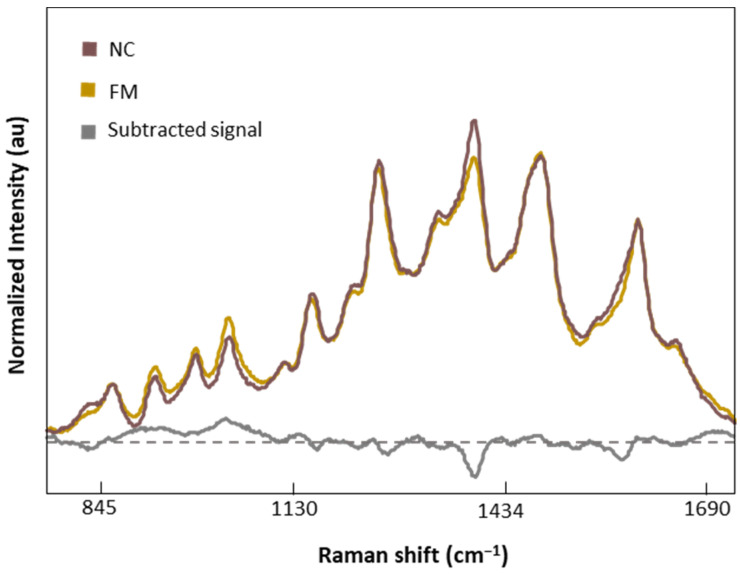
Normalized and averaged SERS spectra of LMF blood samples from FM (*n* = 83) and NC subjects (*n* = 9) and the difference spectrum calculated from the mean SERS spectra among FM and NC subjects.

**Figure 8 biomedicines-12-00133-f008:**
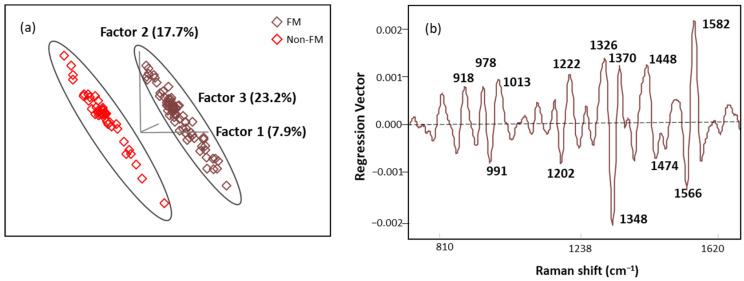
(**a**) The score plot with the first three latent variables (LVs) of the OPLS-DA regression calibration model obtained from the SERS spectral data; (**b**) regression vector plot of OPLS-DA regression calibration model.

**Figure 9 biomedicines-12-00133-f009:**
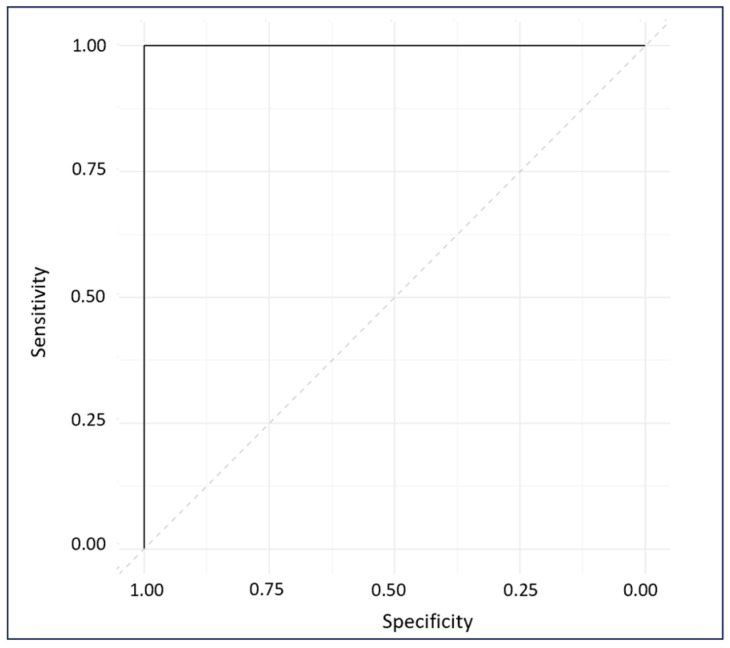
ROC diagnostic plot of external predictions for cross-validated and externally validated sample data.

**Table 1 biomedicines-12-00133-t001:** Detailed distribution of each disorder in the calibration and external validation dataset.

Dataset	FM	SLE	OA	RA	CLBP
Calibration	68	13	10	16	2
Validation	15	5	4	4	0
Total	83	54			

**Table 2 biomedicines-12-00133-t002:** Clinical characteristics of all subjects. Values expressed as mean ± sd; N = number of subjects, age (range). FM: fibromyalgia,. BMI: body mass index. [/] = percentage male/female. CSI: Central Sensitization Inventory. FIQR: Revised Fibromyalgia Impact Questionnaire. SIQR: Revised Symptom Impact Questionnaire. BDI: Beck Depression Inventory. MPI: McGill Pain Index, NC: healthy controls.

	Age	N(M/F) [%M/%F]	BMI	CSI	SIQR	FIQR	MPI	BDI
FM	42.2 ± 14.1	83(7/76) [8/92]	30.9 ± 8.3	64.6 ± 15.1		54.6 ± 21.4	100.3 ± 48.9	23.5 ± 11.1
Non-FM	52.2 ± 16.4	54(10/44) [18.5/81.5]	31.2 ± 12.8	35.4 ± 16.3	33.5 ± 23.0		44.3 ± 39.3	9.5 ± 8.6
NC	45.0 ± 17.7	9(5/4) [55/45]	25.6 ± 4.5	15.8 ± 12.8	4.3 ± 7.0		5.3 ± 8.9	0.7 ± 0.8

**Table 3 biomedicines-12-00133-t003:** Sub-analysis of subject groups. FM: Fibromyalgia, Non-FM: Rheumatoid Arthritis, Systemic Lupus Erythematosus, Osteoarthritis, Chronic Low Back Pain. NC: Healthy Controls. Values expressed as mean ± sd.

	Age	N(M/F) [%M/%F]	BMI	CSI	SIQR	MPI	BDI
RA	51.44 ± 15.55	23(4/19) [17.4/82.6]	29.1 ± 9.7	35.4 ± 14.4	37.3 ± 23.8	42.3 ± 34.6	9.4 ± 7.4
SLE	43.67 ± 15.3	17(2/15) [11.8/88.2]	33.65 ± 9.6	33.27 ± 19.6	31.2 ± 25.5	51.1 ± 53.5	10.4 ± 11.0
OA	67.1 ± 8.9	12(3/9) [25/75]	30.7 ± 10.2	36.8 ± 15.5	25.4 ± 12.2	54.6 ± 36.6	7.3 ± 5.4
LBP	59.5	2(1/1) [50/50]	53.4	53	65.2	80	15
NC	45.0 ± 17.7	9(5/2) [67/33]	25.6 ± 4.5	15.8 ± 2.8	4.3 ± 7.0	5.3 ± 8.9	0.7 ± 0.8

**Table 4 biomedicines-12-00133-t004:** Comparison of FM versus non-FM. Age, BMI, CSI, FIQR/SIQR, MPI, BDI. *p*-values represent two-tailed comparisons between groups.

	FM vs. Non-FM/*p*-Value
Age	<0.001
BMI	0.328
CSI	<0.001
FIQR/SIQR	<0.001
MPI	<0.001
BDI	<0.001

**Table 5 biomedicines-12-00133-t005:** Band assignments of major bands in SERS spectra.

Band (cm^−1^)	Mode	Contributions	Reference
911	C-C stretching	Lys	[47]
967	C-C Stretching	Phe	[48]
1014	Benzene ring breathing	Trp	[49]
1129	C-H bending	Trp and Phe	[50,51]
1189	C-H	Tyr and Phe	[48]
1222	C-H stretching	Phe, Tyr, and Amide III	[50]
1305		Amide III	[50]
1354	C-H bending	Trp	[52]
1453	CH_2_, CH_3_ bending	Phospholipids and lipids	[48,51]
1586	C=C	Phe and Tyr	[51]
1633	Beta sheet	Amide I	[10,51]

**Table 6 biomedicines-12-00133-t006:** Performance of calibration model and external validation set of OPLS-DA model.

Figure of Merit	Calibration Model (*n* = 109)	Prediction Set (*n* = 28)
SECV/SEP	0.02	0.05
R^2^	0.99	1.00
Accuracy%	100	100
Specificity%	100	100
Sensitivity%	100	100

**Table 7 biomedicines-12-00133-t007:** Tentative assignments of the Raman bands in the regression vector of the OPLS-DA algorithm created with SERS spectral data.

Raman Band (cm^−1^)	Mode	Assignment
918	C-C backbone	Ser [53]
978	OCH_3_ stretching	Polysaccharides [10]
991	C–H bending	Phe [54]
1013	C–H bending	Trp [51,54]
1202	Amide III	Phe, Trp, and Amide III [47,51]
1222	C-H stretching	Phe and Tyr [54]
1326	CH_2_ twisting	D-Ser [53]
1348	C-H bending	Trp [52]
1370	C-C stretch	Trp [54]
1448	CH_2_, CH_3_ bending	Phospholipids and lipids [10,48]
1474	C-N stretching	Aromatic ring [47]
1566	C=C bending	Phe [55]
1582	C=C bending	Phe and Tyr [48,51]

## Data Availability

The data presented in this study are available on request from the corresponding author. The data are not publicly available due to privacy concerns.

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
