# Peer review of "Early Diagnosis of Fibromyalgia Using Surface-Enhanced Raman Spectroscopy Combined with Chemometrics"

_biomedicines, 2024, doi:10.3390/biomedicines12010133_

Round 1
Reviewer 1 Report
Comments and Suggestions for Authors
This study is intriguing, and I am inclined to support its publication due to its notable clinical impact. However, I have a few suggestions for improvement:
-
The introduction could benefit from a more comprehensive discussion of the prevalence of fibromyalgia across diverse populations.
-
Elaborating on the hypothesis in the introduction would enhance the clarity and coherence of the study.
-
Could you please confirm whether the study has been registered in any open registry? Clarification on this point is essential.
-
A flow diagram illustrating the entire study process would be a valuable addition to enhance understanding.
-
One concern is the relatively limited recruitment of healthy controls in the study. Providing an explanation for this choice would strengthen the study's transparency and reliability.
I hope these suggestions contribute to the refinement of your study.
Author Response
Thank you for your review of our manuscript. We particularly would like to thank the reviewers for their thoughtful critiques. We have amended the manuscript accordingly and have highlighted those changes below.All changes are in the track changes format for the uploaded manuscript.
Reviewer 1
Comments and Suggestions for Authors
This study is intriguing, and I am inclined to support its publication due to its notable clinical impact. However, I have a few suggestions for improvement:
- The introduction could benefit from a more comprehensive discussion of the prevalence of fibromyalgia across diverse populations.
We agree with the reviewer, and we have added the following information in the Introduction section (lines 45 to 51) “Estimates of the prevalence of FM have been made in various settings, multiple countries and across five continents: Africa, North and South America, Asia, and Europe. The global mean prevalence of FM was 2.7 %, with estimates ranging from a low of 0.4 % in Greece to a high of 9.3 % in Tunisia. In North and South America, the average rate is approximately 3.1 %, 2.5 % in Europe, and 1.7 % in Asia. Prevalence rates globally in women and men are 4.2 % and 1.4 % respectively with a fe-male-to-male ratio of 3:1.” Lines
- Elaborating on the hypothesis in the introduction would enhance the clarity and coherence of the study.
We appreciate the reviewer comment and we have improved our study objective by adding the following sentences in the Introduction section (lines 97 to 100) “Thus, we hypothesize that vibrational spectroscopy may provide a powerful tool for differentiating FM from other disorders. The additional capabilities of SERS spectroscopy may add multifold capabilities to this detection ability. These studies may lead to evidence of prospective therapeutic targets for FM-associated pain.”
- Could you please confirm whether the study has been registered in any open registry? Clarification on this point is essential.
In this study, there were no medical interventions and therefore, we did not meet the accepted National Institutes of Health criteria for a clinical trial. We have added this information in the Material and methods section (lines 110 to 114) “Clinical registries focus on cataloguing the efficacy of medical interventions in clinical trials to achieve health endpoints. In that there were no medical interventions conducted during this study, it did not meet the accepted National Institutes of Health criteria for a clinical trial. Thus, it is not included in an open registry.”
- A flow diagram illustrating the entire study process would be a valuable addition to enhance understanding.
We agree with the reviewer. We have added two figures in the manuscript, Figure 1 that shows the inclusion of FM, Non-FM (RA, SLE, OA and CLBD) and NC patients used for this research and the Figure S4 (supplementary data) where we explain in detail the entire study process.
- One concern is the relatively limited recruitment of healthy controls in the study. Providing an explanation for this choice would strengthen the study's transparency and reliability.
The relatively limited recruitment of healthy controls was due to the target application of the test in rheumatological clinics. These clinics serves a population of patients that experience joint pain and stiffness resulting from rheumatoid arthritis, osteoarthritis, osteoporosis, fibromyalgia, and other related conditions. Thus, healthy subjects would rarely seek care at a rheumatology clinic. Nevertheless, we are currently working on increasing the number of healthy patients. So far, all SERS spectra of healthy controls analyzed have shown distinct patterns when compared to FM subjects, and we have shown it in this paper. We are planning to increase the numbers of subjects analyzed and keep monitoring the SERS spectral pattern for healthy control subjects.
- I hope these suggestions contribute to the refinement of your study.
We really appreciate the comments from reviewer 1. We have improved the overall quality of our manuscript.
Reviewer 2 Report
Comments and Suggestions for Authors
This work reports the use of surface-enhanced Raman spectroscopy with gold nanoparticles to diagnose fibromyalgia and other rheumatic diseases. The authors have demonstrated the feasibility, specificity and accuracy of this approach. The manuscript can be accepted for publication after minor revisions.
1. I find that at least tables 5 and 6 are not correctly cited in the text. I guess that table 4 in line 334 should be table 5 and table 5 in line 338 should be table 6. Other citations of table also need to be checked.
2. It is unclear why references are included in table 5.
3. Line 173, it is more rigorous to indicate centrifugation force.
Comments on the Quality of English LanguageThere are some typos in the text.
Author Response
Comments and Suggestions for Authors
This work reports the use of surface-enhanced Raman spectroscopy with gold nanoparticles to diagnose fibromyalgia and other rheumatic diseases. The authors have demonstrated the feasibility, specificity and accuracy of this approach. The manuscript can be accepted for publication after minor revisions.
- I find that at least tables 5 and 6 are not correctly cited in the text. I guess that table 4 in line 334 should be table 5 and table 5 in line 338 should be table 6. Other citations of table also need to be checked.
We agree with the reviewer, we have correctly cited the tables and figures in the manuscript.
- It is unclear why references are included in table 5.
The Raman bands are linked to functional groups, and it is important to cite other studies that have found the same associations.
- Line 173, it is more rigorous to indicate centrifugation force.
The centrifugation force is explained in line 184 “at 4,000 rpm for 10 min at 4 °C”
- There are some typos in the text.
We agree with the reviewer. We have checked the manuscript to look for typos and some corrections have been made.
Reviewer 3 Report
Comments and Suggestions for Authors
Fibromyalgia is frequently associated with other rheumatic diseases. Nowadays, there are no reliable diagnostic tests, thus history intake coupled with clinical evaluative descriptions are used to identify affected individuals. Physicians are in need of a rapid diagnostic method to easily discriminate between fibromyalgia and other rheumatic conditions.
Even if it is a first attempt to investigate the potential of using SERS in low-molecular-weight-fraction of the human plasma proteome to diagnose FM and related rheumatologic disorders, the results are encouraging and fully justify the continuation of research in this direction.
1. Point 2.1. Patient Sample Recruitment and Sample Storage:
The patients involved in the study are men or women? Please add.In the study you included 54 patients non-FM. How many with RA, SLE, OA and CLBP? Please add.
2. Please add the Au NPs characterization in detail.
3. Typos errors: line 115
Comments on the Quality of English LanguageMinor editing of English language is required
Author Response
Comments and Suggestions for Authors
Fibromyalgia is frequently associated with other rheumatic diseases. Nowadays, there are no reliable diagnostic tests, thus history intake coupled with clinical evaluative descriptions are used to identify affected individuals. Physicians are in need of a rapid diagnostic method to easily discriminate between fibromyalgia and other rheumatic conditions.
Even if it is a first attempt to investigate the potential of using SERS in low-molecular-weight-fraction of the human plasma proteome to diagnose FM and related rheumatologic disorders, the results are encouraging and fully justify the continuation of research in this direction.
- Point 2.1. Patient Sample Recruitment and Sample Storage: The patients involved in the study are men or women? Please add.
The males and female subject numbers are displayed in Table 2 (Results section) and are also in the text in section 3 under clinical characteristics.
- In the study you included 54 patients non-FM. How many with RA, SLE, OA and CLBP? Please add.
We have this information in the Table 1, we have included in this study 18 SLE, 14 OA, 20 RA and 2 CLBD.
- Please add the Au NPs characterization in detail.
We have added Figures S2 and S3 in the supplementary data showing the UV–Visible absorption spectroscopy and dynamic light scattering (DLS) results of AuNPs batch used for this research.
- Typos errors: line 115.
We appreciate the reviewer comment. We have removed the word “with”.
- Minor editing of English language is required.
We agree with the reviewer. We have checked the manuscript to look for typos and some corrections have been made.